

# Two new $^{222}$Rn emanation sources – a comparison study

Tanita J. Ballé[1], Stefan Röttger[1], Florian Mertes[1], Anja Honig[1], Petr Kovar[2], Petr P. S. Otáhal[3], Annette Röttger[1]

[1] Physikalisch-Technische Bundesanstalt (PTB), Bundesallee 100, 38116 Braunschweig, Germany
[2] Czech Metrological Institute (CMI), Regional Branch Prague, Radiová la, 102 00 Prague
[3] National Institute for Nuclear, Biological and Chemical protection (SUJCHBO, v.v.i), 262 31 Milin, Czech Republic

*Correspondence to*: Tanita J. Ballé (Tanita.balle@ptb.de)

**Abstract.** More than 50 % of natural occurring radiation exposure of the general public is due to the noble gas radon ($^{222}$Rn)

and its progenies, causing considerable health risks. Therefore, the European Union has implemented council directive 2013/59/EURATOM to measure $^{222}$Rn activity concentrations and to identify Radon Priority Areas (RPAs) to specify areas where countermeasures are most needed. Although $^{222}$Rn measurements are far spread across Europe, traceability to the international system of units (SI) is still lacking. Consequently, measurement results cannot be reliably compared to each other. The EMPIR project 19ENV01 traceRadon aims to address this issue and has developed two new $^{222}$Rn emanation

sources, intended to be used as calibration standards for reference instruments. The goal of this paper is to investigate and compare the two sources to ensure their quality by comparing the calibration factors estimated from both sources for the same reference instrument. This was done for three reference instruments in total at two experimental sites. Differences of calibration factors for one reference instrument of up to 0.07 were derived. Despite the small differences between the calibration factors, all uncertainties are well within the aspired target uncertainty of 10 % for $k = 1$.

**1 Introduction**

The radioactive noble gas radon ($^{222}$Rn) has piqued the interest of researchers for quite some time due to its impact on natural radiation exposure of the general public and the associated lung cancer risks (Jacobi, 1993). $^{222}$Rn is generated through α-decay of radium ($^{226}$Ra) and part of the $^{238}$U-decay chain. A multitude of Rn isotopes are known to exist, the most abundant being $^{222}$Rn, with a half-life of $T_{1/2} \approx 3.8$ d.

Approximately 3 % to 14 % of all lung cancer deaths are attributed to the exposure of radiation from $^{222}$Rn (progenies), depending on the activity concentration of $^{222}$Rn in a certain area. Therefore, $^{222}$Rn progenies are the second biggest cause for lung cancer after smoking. The World Health Organization (WHO) and other national and international organizations recommend $^{222}$Rn measurements to identify areas with high $^{222}$Rn activity concentrations, so called Radon Priority Areas (RPAs) (Cinelli et al., 2018). Additionally, the identification of RPAs is a major aim of the EU EMPIR project 19ENV01

"Radon metrology for use in climate change observation and radiation protection at the environmental level". The project



results will be implemented to identify RPAs, which is required by European Council Directive 2013/59/EURATOM, which in turn will help decision makers to enforce the respective $^{222}$Rn action plans of the EU member states and improve radiation protection of the general public (Röttger et al., 2021).

In the framework of the International Carbon Observation System (ICOS) the European commission Joint Research Center
(JRC) has published a map of Europe, presenting indoor $^{222}$Rn measurements as early as 2006 (accessible at https://remap.jrc.ec.europa.eu/ Atlas.aspx#).

All European countries operate automatic gamma dose rate systems and atmospheric radionuclide concentration detectors for environmental radioactive monitoring. The results of this radiological monitoring are exchanged through the European Radiological Data Exchange Platform (EURDEP) as requested by EU legislation (Council decision 87/600/Euratom of
December 1987 on community arrangements for the early exchange of information in the event of radiological emergency ELI; Available at https://eur-lex.europa.eu/legal-content; accessed 21 April 2023).

However, despite extensive research, there is no outdoor $^{222}$Rn activity concentration measurement map published as of yet (April 2023) (Cinelli et al., 2018). This is mainly attributed to the challenges of measuring $^{222}$Rn at the low activity concentrations found in outdoor environments (below 100 Bq·m$^{-3}$), making precise and comparable measurements traceable
to the international system of units (SI) complicated. $^{222}$Rn activity concentration in air depends on a multitude of factors, like the Uranium concentration in soil, the temperature and soil permeability, to name but a few (Čeliković et al., 2022). Different methods are implemented at different measurement sites, making comparisons of existing outdoor $^{222}$Rn activity concentration measurements challenging or impossible (Schmithüsen et al., 2017).

A detailed study of measurement devices proved their principle capability of measuring $^{222}$Rn activity concentrations below
100 Bq·m$^{-3}$, but due to the lack of suitable calibration all of them had uncertainties of at least 15 % below 100 Bq·m$^{-3}$ (Radulescu et al., 2022), and therefore no traceability to SI.

Within the EU 40 countries are currently gathering gamma dose rate data at 5500 automated observation stations (data available at https://remap.jrc.ec.europa.eu/Advanced.aspx). The EMPIR project 19ENV01 traceRadon aims to improve this Network by addressing the issues mentioned above and provide outdoor $^{222}$Rn activity concentrations from 1 Bq·m$^{-3}$ to 100
Bq·m$^{-3}$ traceable to SI with uncertainties below 10 % for $k = 1$.

One of the implemented methods to reach this goal is presented in this paper: Two new $^{222}$Rn emanation sources were developed: the Integrated Radon Source Detector (IRSD) and the source developed by the Czech Metrology Institute (in the following referred to as CMI-source). The IRSD represents a completely new class of $^{222}$Rn emanation sources. A layer of $^{226}$Ra is placed directly on top of a commercially available passivated ion-implanted planar silicon semiconductor (PIPS). As
the PIPS-detector is capable of spectrometric measurements of α-particles the emanation of $^{222}$Rn during an experiment can be observed quasi online as described in more detail in section 2.2. The CMI source on the other hand is based on the build-up of $^{222}$Rn within the source and a subsequent dilution with air. Up to a certain point the $^{222}$Rn emanation can be adjusted by variation of the air flowing through the source. Details on the setup of the sources can be found in references (Mertes et al., 2022) and (Fialova et al., 2020). Here we present a comparison study of the two sources regarding their suitability to be





implemented as calibration standards. Thus, a calibration of existing measurement devices at 1 Bq·m⁻³ to 100 Bq·m⁻³ will be possible with the required uncertainty and traceability to SI.

To ensure the quality of the two sources both were investigated with regard to inferred calibration factors at two experimental sites, at Physikalisch-Technische Bundesanstalt (PTB, Germany) and at the National Institute for Nuclear, Biological and Chemical protection (SUJCHBO, v.v.i.; Czech Republic). At PTB a setup under laboratory conditions in a

50 L and a 500 L closed reference volume was chosen. To ensure comparability, a similar setup was chosen at SUJCHBO with a 324 L closed reference volume. In addition, at SUJCHBO a calibration factor was determined with a different experimental setup under outdoor conditions. The comparison is meant to show the reproducibility of calibrations factors regardless of the implemented source and details of the experimental setup. This is seen as an indication of the high quality of both sources.

In Section 2 the results from PTB will be described while Section 3 covers the results obtained at SUJCHBO. In Section 4 the results of both experimental sides will be compared followed by a short summary in Section 5.

## 2 Measurements at PTB

In this section the results of the measurements at PTB, implementing the two new $^{222}$Rn emanation sources as calibration standards, will be described and discussed.

### 2.1 Setup

Both, the IRSD and CMI-source, were measured using radon monitors as reference instruments (Radon Reference Instrument, RRI #1 and RRI #2, both of the AlphaGUARD EF type). In case of the IRSD two different sources of the same type (IRSD #1 and IRSD #2) were implemented in two independent measurements. For the first measurement the IRSD #1 was connected to a 500 L closed reference volume through a standard vacuum KF40 flange T-piece with the RRI #1 placed

inside the reference volume. The RRI #1 was operated in diffusion mode. After a measurement period of about 2 months the IRSD #1 was removed, and the CMI-source placed inside the reference volume with the valves open. Both experiments were repeated in a 50 L closed reference volume, implementing a second RRI and a second IRSD (RRI #2 and IRSD #2).

Comparison of both sources was carried out based on the derived values of the RRI calibration factors $k$ with respect to the certified activity and emanation rate of each source. In the ideal case both calibration factors determined for one RRI will be

identical, as both sources are meant to be suitable as calibration standards and should therefore yield the same calibration factor for the same RRI. For the CMI-source the activity and emanation factor were taken from the issued calibration certificate of CMI (see reference (Grexová et al., 2021)), whereas the PTB development, the IRSD, allows for quasi online, data-driven computation of the $^{222}$Rn activity concentration as described in the following section.



## 2.2 Methods implemented for the Integrated Radon Source Detector (IRSD) at PTB

In the following the method used to derive the $^{222}$Rn activity concentration will be outlined. First, the activity of $^{222}$Rn remaining in the $^{226}$Ra-source, $A_{\mathrm{Rn}}^{\mathrm{s}}$, can be calculated according to

$$\frac{\mathrm{d}A_{\mathrm{Rn}}^{\mathrm{s}}}{\mathrm{d}t} = -\lambda_{\mathrm{Rn}}A_{\mathrm{Rn}}^{\mathrm{s}} + \lambda_{\mathrm{Rn}}A_{\mathrm{Ra}}^{\mathrm{s}} - \lambda_{\mathrm{Rn}}\eta(t). \quad \textbf{(1)}$$

This formula contains the decay constant of $^{222}$Rn, $\lambda_{\mathrm{Rn}}$, the activity of $^{222}$Rn decaying within the $^{226}$Ra-source, $A_{\mathrm{Rn}}^{\mathrm{s}}$ (negative contribution), of all $^{222}$Rn produced in the source, $A_{\mathrm{Ra}}^{\mathrm{s}}$ (through α-decay of $^{226}$Ra-atoms; positive contribution) per unit time, and finally of $^{222}$Rn emanated into the gas surrounding the source, $\eta(t)$ (negative contribution), in terms of atoms per unit

time.

Since it is assumed that the reference volume is perfectly hermetically closed against any losses of $^{222}$Rn the activity of $^{222}$Rn evolves by

$$\frac{\mathrm{d}A_{\mathrm{Rn}}^{\mathrm{v}}}{\mathrm{d}t} = -\lambda_{\mathrm{Rn}}A_{\mathrm{Rn}}^{\mathrm{v}} + \lambda_{\mathrm{Rn}}\eta(t) . \qquad \textbf{(2)}$$

Note that the IRSD measures only α-particles emitted from within its layer of $^{226}$Ra. Due to the setup (see reference (Mertes et al., 2022)) the contributions of α-decays from $^{222}$Rn in the reference volume, $A_{\mathrm{Rn}}^{\mathrm{v}}$, are negligible in comparison. Since the

α-decay of $^{226}$Ra and $^{222}$Rn is associated with different α-particle energies $A_{\mathrm{Rn}}^{\mathrm{s}}$ and $A_{\mathrm{Ra}}^{\mathrm{s}}$ can both be determined based on the α-spectra measured by the PIPS detector inside the IRSD. The RRI on the other hand measures solely the activity concentration in the volume, from which $A_{\mathrm{Rn}}^{\mathrm{v}}$ is derived by multiplication with the known reference volume. The evolution of $A_{\mathrm{Rn}}^{\mathrm{v}}$ is shown in equation (2). It is linked to $A_{\mathrm{Rn}}^{\mathrm{s}}$ and $A_{\mathrm{Ra}}^{\mathrm{s}}$ through $\eta(t)$, as can be seen from comparison of equation (1) and equation (2). $A_{\mathrm{Rn}}^{\mathrm{v}}$ may also be inferred from the dynamics of the build-up of $^{222}$Rn in the volume, the continuity of the total

amount of $^{222}$Rn expressed by equations (1) and (2) and the supporting IRSD measurements. The statistical inference of $A_{\mathrm{Rn}}^{\mathrm{v}}$ based on the IRSD measurements of $A_{\mathrm{Rn}}^{\mathrm{s}}$ and $A_{\mathrm{Ra}}^{\mathrm{s}}$ will be described in the subsequent outline.

First $A_{\mathrm{Ra}}^{\mathrm{s}}$ follows as:

$$\frac{\mathrm{d}A_{\mathrm{Ra}}^{\mathrm{s}}}{\mathrm{d}t} = -\lambda_{\mathrm{Ra}}A_{\mathrm{Ra}}^{\mathrm{s}} \qquad \textbf{(3)}$$

The coupled ordinary differential equations (ODEs) (1) and (3) may be combined by defining

$$\vec{A} = \begin{pmatrix} A_{\mathrm{Rn}}^{\mathrm{s}} \\ A_{\mathrm{Ra}}^{\mathrm{s}} \end{pmatrix} \quad \text{and} \qquad \vec{L} = \begin{pmatrix} -\lambda_{\mathrm{Rn}} \\ 0 \end{pmatrix}, \qquad \textbf{(3)}$$

as well as the Matrix

$$\overline{K} = \begin{pmatrix} -\lambda_{\mathrm{Rn}} & \lambda_{\mathrm{Rn}} \\ 0 & -\lambda_{\mathrm{Ra}} \end{pmatrix} \qquad \textbf{(4)}$$

which yields the combined inhomogeneous ODE

$$\mathrm{d}\vec{A} = \overline{K}\vec{A}\mathrm{d}t + \vec{L}\eta(t)\mathrm{d}t. \qquad \textbf{(5)}$$

Only $\vec{A}$ and $\eta$ depend on time, but $\overline{K}$ and $\vec{L}$ do not.





This differential equation can be solved by the integrating factor method to yield

$$\vec{A}(t) = e^{\overline{K}(t - t_0)} A_{t_0} + \int_{t_0}^{t} e^{\overline{K}(t - \tau)} \vec{L} \eta(\tau) d\tau. \qquad (7)$$

Since radioactivity is a Poisson-process by definition, noise in the measurement of $\vec{A}(t)$ cannot be avoided, and therefore a mere estimation of the time derivative in equation (1) yields unsatisfactory results in the pursuit of the determination of $\eta(t)$.

On the other hand, no information about $\eta(t)$ can be inferred without relying on data. To model the temporal evolution of $\eta(t)$, it is described as a stochastic process. As a result, it is possible to capture its time-dependent uncertainty. The emanation is modelled to obey the following stochastic differential equation (SDE) in the Itō sense, which has a Gaussian process as a solution:

$$d\eta = \sigma d\beta_t , \qquad (6)$$

where $d\beta_t$ describes the increments of a standard one-dimensional Wiener process and $\sigma$ represents the standard variation.

The model for the emanation can be combined with the dynamics of the $^{222}$Rn-source of the IRSD and the accumulation of $^{222}$Rn within the reference volume (essentially the combination of equations (1) – (3) and (6)), through the definition of a state vector, $\vec{x}$, which yields a combined SDE and may be solved analogous to equation (7) as

$$\vec{x}(t) = \begin{pmatrix} A_{Rn}^{V} \\ A_{Rn}^{s} \\ A_{Ra}^{s} \\ \eta \end{pmatrix}(t) \qquad (7)$$

$$= e^{\overline{F}(t - t_0)} \vec{x}(t_0) + \int_{t_0}^{t} e^{\overline{F}(t - \tau)} \vec{L} \eta(\tau) d\beta_\tau, \qquad (8)$$

with

$$\overline{F} = \begin{pmatrix} -\lambda_{Rn} & 0 & 0 & \lambda_{Rn} \\ 0 & -\lambda_{Rn} & \lambda_{Rn} & -\lambda_{Rn} \\ 0 & 0 & -\lambda_{Ra} & 0 \\ 0 & 0 & 0 & 0 \end{pmatrix} \qquad (9)$$

and

$$\vec{L} = \begin{pmatrix} 0 \\ 0 \\ 0 \\ \sigma \end{pmatrix}. \qquad (10)$$

The process of the IRSD measurements is described as

$$p(\vec{y_t}|\vec{x_t}) \propto \text{Poisson}\left(\overline{H}\vec{x_t}\right) \approx \text{Normal}\left(\overline{H}\vec{x_t}, \overline{H}\vec{x_t}\overline{H}^{\mathrm{T}}\right) \qquad (11)$$



Where $\overrightarrow{y_t}$ signifies a vector of peak-areas corresponding to $^{226}$Ra and the $^{222}$Rn peaks obtained from the IRSD α-spectrum at time $t$, respectively. Therein, a Gaussian approximation was chosen and the components of $\overline{H}$ are known from the calibration of the IRSD as described in reference (Mertes et al., 2022), which is traceable to the primary defined-solid angle (DSA) α-particle spectrometer of PTB. The peak-areas were determined from each IRSD α-particle spectrum using non-linear

regression against a Poisson likelihood also described in reference (Mertes et al., 2022), while neglecting the integrating behavior of the spectrometric measurements.

Inference of the state vector entails computation of the collection of probability density functions $p(\overrightarrow{x_t} \mid \vec{y}_{1,\ldots,T})$, which depend on all collected IRSD spectra within the measurement interval $T$, indicated by the notation "$\vec{y}_{1,\ldots,T}$", and for all desired instants in time $t$. In this case these are the time-instants where the RRI reported a measurement of $A_{Rn}^v$. The computation of

the statistical moments (mean vector and covariance matrix) of $p(\overrightarrow{x_t} \mid \vec{y}_{1,\ldots,T})$ may be achieved by the recursions of the Kalman-Filter and the Rauch-Tung-Striebel smoother for this specific type of model (see references (Särkkä and Solin, 2019; Rauch et al., 1965; Kalman, 1960; Särkkä, 2013)). The matrix exponential required in the discretization of the dynamical system, as given by equation (8), was computed symbolically.

A remaining unknown parameter of this model is the standard deviation $\sigma$ in equation (6). The maximum likelihood

estimator for $\sigma$ was determined by maximizing the marginal log-likelihood of the measurement series $(\vec{y}_{1,\ldots,T})$, which is computed alongside the Kalman-Filter recursions (analogous to references (Rauch et al., 1965; Kalman, 1960)). Since the reference volume is known, the probability density for $A_{Rn}^v$ can be computed at any time instant, depending on the observed IRSD spectra, by implementing the described modelling procedure.

The uncertainty of the inferred emanation increases as the temporal distance to related IRSD measurement time instants

increases, which is a feature of the model definition and captures the fact, that the evolution of $\eta(t)$ is unknown in the absence of IRSD measurements.

The Kalman-Filtering approach requires the specification of a Gaussian prior distribution of the state vector for the time $t_0$. At time $t_0$, marking the beginning of the RRI measurements, the reference volume was opened to obtain a stable initial state. While the actual $^{222}$Rn activity concentration in the reference volume was low at this point, it was assumed to be greater than

zero. To alleviate this, the $^{222}$Rn activity concentration at $t_0$ was determined as the value which maximized the linearity of the RRI response in comparison to the inferred $^{222}$Rn activity concentration evolution at the assumed background reading. The background contribution of the RRI was later determined to $(30 \pm 17)$ Bq·m$^{-3}$, based on measurements without a source while the reference volume was flooded with $^{222}$Rn-free synthetic air.

## 2.3 Results and discussion at PTB of the IRSD

The results of the measurements and the calculations in Bq versus time are shown in Figure 1 and Figure 2. The measurement is shown as grey dots. The smoothing results, based on the IRSD data and treated according to the procedure





described in section 2.2, are presented as a blue line, whereas the shaded blue areas signify the marginal 1 $\sigma$ confidence intervals, with the respective assigned statistical uncerainties.

The measurements of the RRI #1 and RRI #2 after conversion into Bq with the known reference volume, are plotted in the
upper panel of Figure 1 and Figure 2, respectively, and represent an independent measurement of $A_{Rn}^v$. The $^{222}$Rn activity concentration in the reference volume slowly rises, as $^{222}$Rn is released from the source into the reference volume until radioactive equilibrium is reached. The middle and lower panels show the activities of $^{222}$Rn and $^{226}$Ra remaining in the source, $A_{Rn}^s$ and $A_{Ra}^s$, respectively. Shown in the panels are the peak areas determined from the IRSD spectra of $^{222}$Rn and $^{226}$Ra. Note, that the emanation from the source is not stable due to changes in the relative humidity (middle panel), however,
this was considered based on the collected IRSD α-particle spectra, and the modelling procedure. $A_{Ra}^s$ seems constant over the whole measurement (lower panels). This is consistent with the long half-life of $^{226}$Ra of $T_{1/2} \approx 1600$ a, causing any changes in $A_{Ra}^s$ to be negligible on the timescales of these measurements.

Comparing the upper panel of Figure 1 with the one from Figure 2 it is apparent that the standard deviation of RRI #1 is much smaller than the standard deviation of RRI #2 despite both instruments being of the same type. This is due to the
difference in setup: Both instruments measure not the activity of $^{222}$Rn, but the activity concentration. As the volume is known, the activity can be easily calculated. However, as RRI #1 was placed inside a 50 L volume and RRI #2 was placed in a 500 L volume the absolute values measured by the two RRI differed by one order of magnitude resulting in a higher absolute activity concentration and thus smaller standard deviation for the RRI which was placed in the smaller volume.

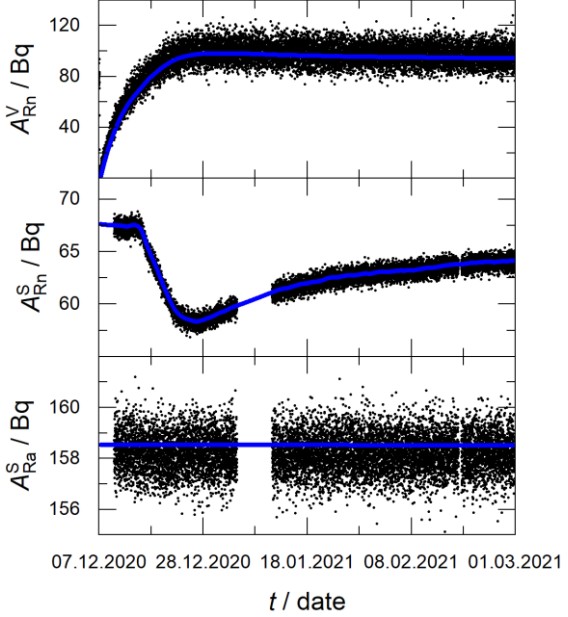

**Figure 1 Measurements of the activity in Bq versus time. The upper panel shows the $^{222}$Rn activity, $A_{Rn}^v$, as measured by the RRI #1. Measurements of the IRSD #1 are shown in the middle and lower panel. Presented are the activities of $^{222}$Rn and $^{226}$Ra, $A_{Rn}^s$ and $A_{Ra}^s$, respectively. In all panels the gray points represent the measurement. In the middle and lower panel, the blue line shows the fit to the data based on equation (8), while in the upper panel the blue line represents the fit result from the lower panels.**



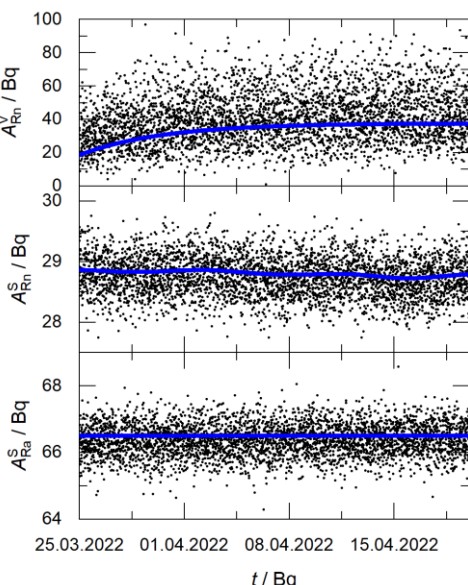

**Figure 2 Activity plotted versus time for RRI #2 and IRSD #2. The gray points represent the measurement while the blue line represents the fit to the data based on equation (8) (middle and lower panel). In the upper panel $A^v_{Rn}$ as measured by the RRI #2 is shown in [Bq] as gray points and the activity derived from equation (10) as the blue line. The middle and lower panel depict the activities $A^s_{Rn}$ and $A^s_{Ra}$, respectively, as measured by the IRSD #2 in Bq.**

The calibration factor, $k$, resulting from comparison of the IRSD data with the respective RRI, is obtained as the reciprocal slope from the (unweighted) linear regression of the indicated RRI $^{222}$Rn activity concentration inferred from IRSD data. For the RRI #1 it is inferred to

$$k = 1.019 \pm 0.015$$

and for RRI #2 it is determined as

$$k = 0.981 \pm 0.015.$$

The uncertainty of the calibration factor is assumed to be 1.5 %, which is based on the systematic uncertainty of the IRSD calibration using the primary defined solid angle α-spectrometry (DSA) standard. Even though the outlined approach allows to determine the statistical uncertainty associated with the IRSD measurements, this contribution is considered negligible in comparison, because of the high number of datapoints. In addition, it is assumed that the influence of the uncertainty in $\sigma$

resulting from the model is negligible.

## 2.4 Methods implemented with the CMI-source at PTB

The source allows to create atmospheres with different $^{222}$Rn activity concentrations, depending on the flow rate of air through the source. At PTB no active air flow was installed. The source was placed in the closed reference volume and $^{222}$Rn diffuses through the open valves into the reference volume.



The intrinsic background of the measurement device in the reference volume without a $^{222}$Rn-source was determined with $^{222}$Rn-free synthetic air to a value of

$$\Delta M_0 = (30 \pm 17) \text{ Bq·m}^{-3}. \tag{12}$$

The following model was implemented to calculate the sensitivity and the calibration factor of the respective RRI:

$$k_c = \frac{1}{k} \quad \text{with} \quad k = \frac{C}{(\Delta M - \Delta M_0)} \tag{13}$$

with

$$\Delta M = \frac{M}{\Delta t} \quad \text{and} \quad \Delta M_0 = \frac{M_0}{\Delta t}. \tag{14}$$

$\Delta M$ represents the measured $^{222}$Rn activity concentration (including background) during time $\Delta t$, while $\Delta M_0$ represents the
background contribution. The reference $^{222}$Rn activity concentration, $C$, is calculated from the $^{226}$Ra activity of the source, $A$, the emanation coefficient, $\chi$, and the reference volume of the vessel, $V$, reduced by the volume occupied by the included components, such as source and monitor. The $^{226}$Ra activity from the source and the emanation coefficient were taken from the calibration certificate (see reference (Grexová et al., 2021)). The reference volume was carefully determined by measuring the volumes of the barrel, of the detector and the source.

Corrections for a background activity concentration, $C_{bg}$, and a loss of activity concentration, $\Delta C$, (in case of leakage) are implemented for the purpose of the uncertainty calculation:

$$C = C_s - C_{bg} - \Delta C \tag{15}$$

with $\quad C_s = \frac{\chi A}{V}.$

The model shows consistency with the assumption that $\Delta C = 0$ and $C_{bg} = 0$, but it is important to note that this assumption is valid.





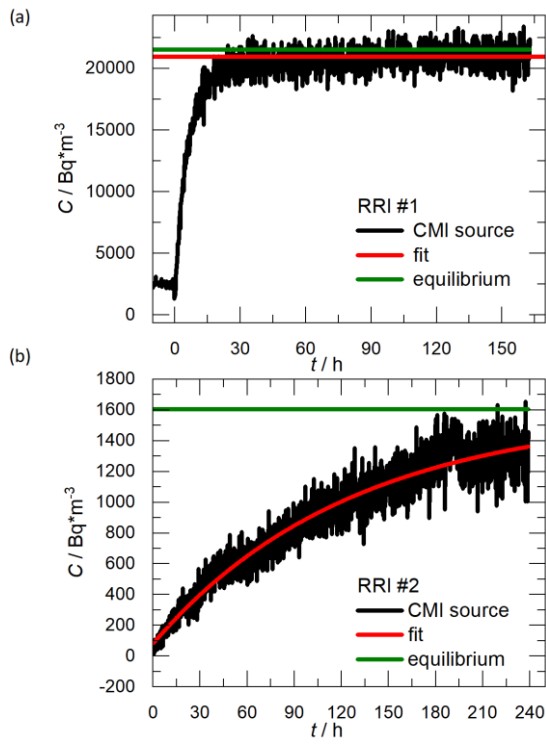

**Figure 3** $^{222}$**Rn activity concentration,** *C***, in Bq·m$^{-3}$ plotted versus time. The measurement is shown in black. The red line is the fit to the data, while the green line represents the** $^{222}$**Rn equilibrium activity concentration. (a) RRI #1 (b) RRI #2.**

## 2.5 Results and discussion at PTB of the CMI- source

The measurements on the CMI-source performed at PTB are shown in Figure 3. The results for the $^{222}$Rn activity concentration in Bq·m$^{-3}$ are plotted as a function of time and represented by the black line. The sudden increase at the beginning marks the opening of the valves of the source. Even before that the $^{222}$Rn activity concentration exceeds the background. This is ascribed to a leakage in the source valves causing some $^{222}$Rn to diffuse into the reference volume, even when the valves are closed.

Once the valves of the source are opened $^{222}$Rn gas, formerly trapped within the volume of the source, is released into the reference volume causing a sudden rise in $^{222}$Rn activity concentration. The fit to the data (red line) was started once this process had finished. Afterwards, the $^{222}$Rn activity concentration continues to rise until radioactive equilibrium is reached. On the timescale of the measurement this is not the case, but the equilibrium $^{222}$Rn activity concentration was calculated as part of the fitting process and is indicated by the green line.

The calibration shown here results in a calibration factor of

$k = 1.056 \pm 0.019$



for RRI #1 and of

$$k = 1.022 \pm 0.017$$

for RRI #2.

The relative humidity, temperature and pressure during the measurement were monitored as well but are not shown, since no significant changes were observed.

## 3 Measurements at SUJCHBO

In this chapter the comparison made at SUJCHBO will be described. First the general data analyses will be outlined, followed by the measurements made with the IRSD. Measurements implementing the CMI-source will follow before the

actual comparison of the sources.

### 3.1 Measurements under laboratory conditions

The laboratory conditions are described in the following and can be found in more detail in reference (Fialova et al., 2020). A newly developed piece of equipment is a part of the Czech primary radon measurement device situated at SUJCHBO, v.v.i. Kamenna (Central Bohemia). In particular, the equipment consists of an airtight Low-Level Radon CHamber

(LLRCH), a humidifier, a mass flow controller of the Bronkhorst® EL-Flow type (Bethlehem—PA, USA), and an aerosol filter. A bottle of synthetic $^{222}$Rn-free air can be attached. To achieve a specific low-level $^{222}$Rn activity concentration, it is necessary to ensure (1) a constant $^{222}$Rn supply and (2) a defined ventilation in the $^{222}$Rn chamber. Because of the location of SUJCHBO, which is close to a former uranium mine, it is possible to measure an outdoor $^{222}$Rn activity concentration in the range of tens or hundreds of Bq·m$^{-3}$. Therefore, it would not be possible to achieve a low-level $^{222}$Rn activity concentration

there without using a bottle with a suitable supply of $^{222}$Rn-free air.

On its way to the low-level $^{222}$Rn-source, air from the bottle with synthetic $^{222}$Rn-free air passes through a protective aerosol particle filter and then the calibrated mass flow controller. After passing through the source the resulting mixture of air and $^{222}$Rn passes through a humidifier to the $^{222}$Rn chamber. The humidifier is included to ensure that the measurement conditions are as realistic as possible. The homogeneity of the atmosphere inside the $^{222}$Rn chamber is ensured by means of a

continually regulated ventilator (the airflow speed can be set in the range of 0.1 m·s$^{-1}$ – 3.5 m·s$^{-1}$). Sensors for the measurement of the climatic conditions are placed inside the LLRCH.

The LLRCH is of cylindrical shape and made of steel with a volume of 324 L. The whole chamber is earthed, and the inner surface is painted with a special coating to minimize the deposition of $^{222}$Rn decay products on the walls. The LLRCH is equipped with four sampling points to which system components can be connected to take samples of the inside air. These

$$C(t) = C_0 \cdot e^{-(\lambda_{Rn}+k)\cdot t} + \frac{R}{V(\lambda_{Rn}+k)}\left(1 - e^{-(\lambda_{Rn}+k)\cdot t}\right) \qquad (16)$$

$$C_{Rn}^V = \frac{R_{Rn}}{Q_{settled} \cdot \frac{M \cdot p_Q/R \cdot T_Q}{M \cdot p_C/R \cdot T_C} + \lambda_{Rn} \cdot V} \qquad (17)$$





points are located in such a way that they allow sampling from different locations of the chamber. The climatic monitoring capability includes temperature and air pressure readings by sensors placed inside and outside the [222]Rn chamber (to monitor the differential pressure between the chamber and the laboratory atmosphere). In addition, the relative humidity inside the [222]Rn chamber is monitored. The airtightness of the LLRCH was verified through a series of experiments as described in reference (Fialova et al., 2020).

The emanation power of [222]Rn from a [226]Ra-source depends on the humidity of the air flowing through the source. Synthetic air is ultra-dried, but to ensure this is the case also after passing through the source a humidifier was placed behind the [222]Rn-source and the relative humidity in the chamber was measured with and without the humidifier being connected. When the humidifier was not connected, the relative humidity in the chamber was very close to zero. In case of the humidifier being connected, the relative humidity in the chamber was in the range of 40 % – 60 % depending on the setting of the humidifier.

**Table 1 Determined IRSD parameters.**

| | |
|---|---|
| Activity [226]Ra | 153.3 (5) Bq |
| Radon emanation power | 0.575 (2) |
| Source emanation ability | 0.18 (1) mBq·s$^{-1}$ |
| Activity [222]Rn | 65.2 (4) Bq |
| Activity [218]Po | 61.3 (3) Bq |
| Activity [214]Po | 60.9 (2) Bq |

**Table 2 CMI-source parameters as specified in reference** (Grexová et al., 2021)**.**

| | | |
|---|---|---|
| Activity [226]Ra | 1136 (17) Bq | |
| Radon emanation power | 0.9552 (19) | 280 |
| Source emanation ability | 2.3 (1) mBq·s$^{-1}$ | |

**3.2 Measurements under field conditions**

For better comparison a similar setup was chosen for the measurements under field conditions. The respective source was
connected to an AlphaGUARD (RRI #3) and measured in flow-through mode. In addition, a second AlphaGUARD (RRI #4) was implemented for the purpose of background measurements. With that, the high [222]Rn activity concentrations of the outdoor air mentioned above were taken into account. The measurement procedure consisted of three phases: During the first phase both RRIs measured the air flow without the [222]Rn-source. Consequently, both should measure the same (outdoor) [222]Rn activity concentration. At the second phase RRI #4 remained connected to the [222]Rn-source (unchanged compared to
the first phase), but RRI #3 was connected to the [222]Rn-source. In the third Phase again both RRI were not connected to the [222]Rn-source (analog to the first phase) and, based on the comparison of the measurements of RRI #3 and RRI #4, it was possible to determine the outdoor [222]Rn activity concentration which would be measured in diffusion mode.



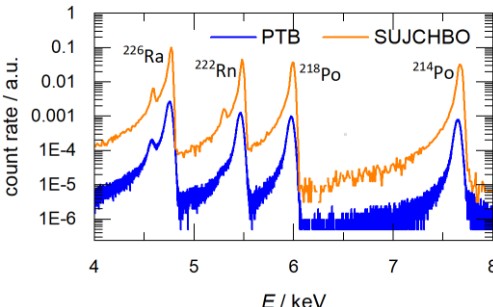

**Figure 4 Comparison of α-spectrums of the IRSD. The orange line was measured at SUJCHBO, while the blue line was measured at PTB.**

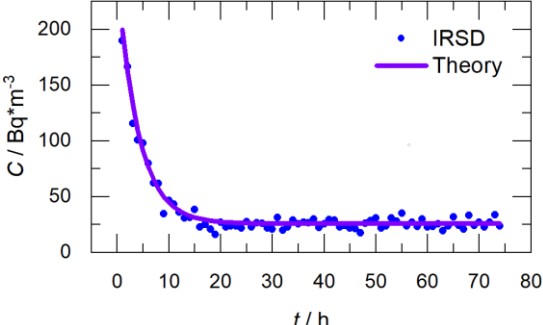

**Figure 5 $^{222}$Rn activity concentration produced by the IRSD versus time (blue dots) including a fit based on the radioactive decay law (purple line).**

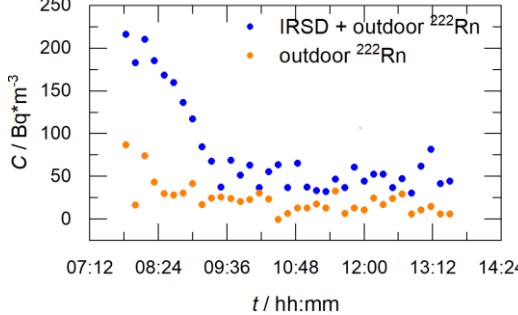

**Figure 6 $^{222}$Rn activity concentration of the IRSD as measured by RRI #3 (blue dots) and RRI #4 (orange dots) versus time.**





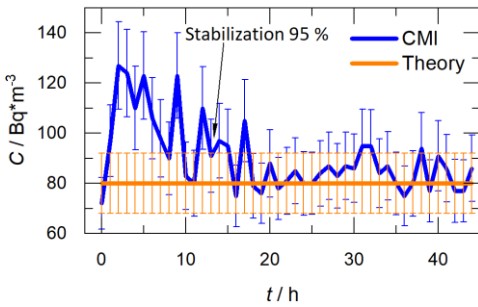

**Figure 7 ²²²Rn activity concentration created by the CMI-source under laboratory conditions. The reference value of 80 Bq·m⁻³ (red) was calculated according to equation (19).**

## 3.3 Reference level of radon for the CMI-source

During the equipment design, a model of constant ²²²Rn input and constant ventilation was applied for the CMI-source as quantified in equation (18). Where $C$ is the ²²²Rn activity concentration at a time $t$; $\lambda_{Rn}$ is the decay constant of ²²²Rn; $k$ is the air exchange intensity and $V$ is the reference volume of the ²²²Rn chamber.

For steady state ($t = \infty$) with a constant air exchange intensity and constant ²²²Rn activity concentration, equation (19) applies. Where $Q_{settled}$ is the flow rate; $M$ is the molar mass; $p_Q$ is the air pressure at the time of the calibration (1013,25 hPa); R is the molar gas constant $T_Q$ is the temperature at the calibration (273.16 K); $p_C$ is the measured air pressure during the experiment; $T_C$ is the measured temperature during the experiment; and $R_{Rn}$ is the ²²²Rn emanation power.

Note, that $C_{Rn}^V$ only depends on the flow rate $Q_s$. All other parameters were monitored and turned out to be constant during the measurement.

## 3.4 Integrated Radon Source Detector (IRSD)

The Alpha Spectrometer Model 7401 was used to determine the ²²⁶Ra activity, ²²²Rn emanation power and source emanation ability of the IRSD. Two α-spectra, measured at SUJCHBO and PTB, respectively, are compared in Figure 4. Based on the results from α-spectrometry processing, the parameters as specified in table 1 of the supplied IRSD were determined.

For the measurements the IRSD was placed in a flow-through flask and connected to the LLRCH. An AlphaGUARD was operated in diffusion mode. A background ²²²Rn activity concentration of the AlphaGUARD was determined as ($2.42 \pm 0.06$ Bq·m⁻³) and subsequently subtracted from the results.

The implemented measurements and evaluation of their results lead to a calibration factor of

$$k = 0.88 \pm 0.04.$$



A large part of the determined uncertainty is formed by the uncertainty associated with the determination of the $^{222}$Rn activity concentration by the AlphaGUARD. The stated uncertainty applies to $k = 1$. It is higher than the uncertainty determined at PTB due to the PIPS-detector within the IRSD not being used. Since the $^{222}$Rn emanation is highly dependent on humidity no outdoor measurements were performed.

**3.5 CMI-source**

The main parameters of the CMI-source were taken from the delivered certificate (see reference (Grexová et al., 2021)) and are summarized in table 2. A flowrate of 1.74 L·min$^{-1}$ through the CMI-source was used to achieve a $^{222}$Rn activity concentration of 80 Bq·m$^{-3}$ in accordance with equation (**17**). The stabilization time required to reach the desired $^{222}$Rn activity concentrations in the LLRCH was estimated at 20 hours. The course of the experiment is shown in Figure 7.

The implemented measurements and evaluation of their results lead to a calibration factor of

$$k = 0.95 \pm 0.01.$$

A large part of the determined uncertainty is formed by the uncertainty associated with the determination of the $^{222}$Rn activity concentration by the AlphaGUARD. The stated uncertainty applies to $k = 1$.

During the field experiments, either one or two RRI (RRI #3 and RRI #4) were used to measure the outdoor $^{222}$Rn activity
concentration in three distinctive phases as described in section 3.2. The RRI #3 connected to the CMI-source was operated in flow-through mode. Figure 8 presents the results of this approach.

To determine the required value of the $^{222}$Rn activity concentration of the connected CMI-source (blue dots), it is necessary to subtract the values of the $^{222}$Rn activity concentration in the outdoor air (green dashed line in Figure 8). In the case of determining the $^{222}$Rn activity concentration in the outdoor air with the help of RRI #3, it is necessary to set aside two values
(at a ten-minute sampling interval) after disconnecting the source. These two values represent the $^{222}$Rn decay products that were deposited in the RRI #3's chamber and increase the background of the instrument.

Calibration factors determined using the CMI-source in the field and one or both RRI were determined as follows:

$$k = 1.13 \pm 0.14$$

for both RRI and

$$k = 1.15 \pm 0.14$$

for one RRI.



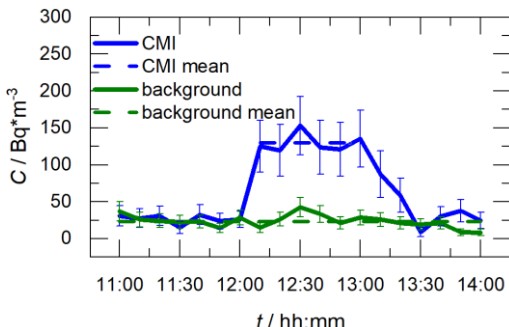

**Figure 8 $^{222}$Rn activity concentration versus time implementing the CMI-source under field conditions with RRI #3 (blue) and RRI #4 (green).**

**Table 3 Calibration factors, _k_, determined for two Radon Reference Instruments (RRI #1 and RRI #2) with both sources at PTB and equilibrium activity concentrations, _C_, in Bq·m$^{-3}$ of the respective measurements.**

| PTB | PTB IRSD system | | CMI source | |
|---|---|---|---|---|
| | _k_ | _C_ [Bq·m$^{-3}$] | _k_ | _C_ [Bq·m$^{-3}$] |
| RRI #1 | 1.019 ± 0.015 | 1925 | 1.056 ± 0.019 | 21547 |
| RRI #2 | 0.981 ± 0.015 | 56.3 | 1.022 ± 0.017 | 1605 |

**Table 4 Calibration factors, _k_, determined for a Radon Reference Instrument (RRI) with both sources at SUJCHBO and equilibrium activity concentrations, _C_, in Bq·m$^{-3}$ of the respective measurements. Note, that the determined uncertainty of the IRSD is higher compared to PTB, because the detector within the IRSD was not used.**

| SUJCHBO | PTB IRSD system | | CMI source | |
|---|---|---|---|---|
| | _k_ | _C_ [Bq·m$^{-3}$] | _k_ | _C_ [Bq·m$^{-3}$] |
| RRI laboratory conditions | 0.88 ± 0.04 | 22.8 | 0.95 ± 0.01 | 80 |
| RRI outdoor conditions | - | - | 1.13 ± 0.14 | 129.8 |



## 4 Comparison of PTB and SUJCHBO

The measurements prove both sources to be capable of providing stable reference atmospheres below 100 Bq·m$^{-3}$. The derived calibration factors at SUJCHBO are summarized in table 4. Even when the same device was implemented significant differences can be observed. The uncertainty of the calibration factor from the RRI determined by implementing the IRSD is higher than obtained by implementing the CMI-source. This is in contrast to the results from PTB (see table 3). The reason for that is due to the PIPS-detector within the IRSD not being used and as a result the $^{222}$Rn activity concentration is not well defined. Furthermore, the uncertainties of the calibration factors inferred at PTB and SUJCHBO are higher when a lower $^{222}$Rn activity concentration was measured.

Differences in the calibration factor determined with each of the two sources, respectively, for the same RRI are mainly attributed to fit uncertainties and the very different methods used in the creation of the reference $^{222}$Rn activity concentration by the two sources: The CMI-source causes a high $^{222}$Rn activity concentration in a small volume within the source that is diluted for the calibration in a low-level atmosphere and requires constant emanation of $^{222}$Rn (realized by constant environmental parameters). The IRSD, on the other hand, directly creates a low-level reference $^{222}$Rn activity concentration in an atmosphere and does not require constant environmental parameters, as the $^{222}$Rn emanation can be determined at a ten-minute interval quasi online.

All calibration factors determined are close to 1 indicating the high quality of the RRI. Furthermore, all procedures result in an uncertainty of the calibration factors smaller than 10 %, which was the aspired goal.

## 5 Conclusions

The two $^{222}$Rn-sources were carefully analyzed and compared at 2 experimental sites (SUJCHBO and PTB), to determine their suitability as standard calibration radon ($^{222}$Rn) sources. Although both sources were thoroughly characterized the measurements result in differing calibration factors for the same reference instrument. Nonetheless, they are well within the aspired goal of an uncertainty of 10 % for $k = 1$. The comparison of the two sources proved that they are both of high quality. The next step is to implement the new calibration sources, possibly for the calibration of the new transfer standards developed in the same project.





## Author contribution

Stefan Röttger, Florian Mertes, Anja Honig and Annette Röttger designed and executed the experiments at PTB as well as data analysis. The CMI-source was prepared by Petr Kovar. Experiment design, execution and data analysis at SUJCHBO was carried out by Petr Otahal. Tanita Ballé prepared the mamuscript with the help of all co-authors.

## Competing interests

The authors declare that they have no conflict of interest.

## Special issue statement

This paper was prepared to be part of the Special issue "Outcomes of the traceRadon project: radon metrology for use in climate change observation and radiation protection at the environmental level"

## Acknowledgements.

This project 19ENV01 traceRadon has received funding from the European Union's Horizon 2020 research and innovation
program. 19ENV01 traceRadon denotes the EMPIR project reference.

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
