# Peer review of "Two new $^{222}\text{Rn}$ emanation sources – a comparison study"

_EGUsphere, 2023_

## Referee Comment (RC1)

The study "**Two new 222Rn emanation sources – a comparison study" by Balle et al.** presents a comparison of two new $^{222}$Rn emanation sources using identical reference instrument at distinct sites: PTB and CMI. The findings revealed varying calibration factors, although they fall well within the targeted uncertainty of 10% (k=1). These sources play a crucial role in addressing a significant gap in traceability for both the radiation protection and climate monitoring communities. They demonstrate the fundamental capability to measure $^{222}$Rn activity concentrations below the 100 Bq m$^{-3}$, with uncertainty of 10% (k=1). Achieving these uncertainties contributes on establishment of traceability to the International System of Units (SI).

**My comments are below:**

**Introduction:** *The introduction is well-written; however, I recommend emphasizing the significance of these two sources for both the radiation protection and climate communities. While the importance in terms of radiation protection is evident, it's essential to also clarify its relevance to the climate community, which is currently missing.*

*Also, it is missing a short description or reference on current calibration sources.*

**Line 45:** "$^{222}$Rn activity concentration in air depends on a multitude of factors, like the Uranium concentration in soil, the temperature and soil permeability…" – *I would say that atmospheric radon concentration depends mostly on the atmospheric process (ref: Chambers et al., Kikaj et al.,)*

**Line 41-48:** *I would restructure this section to emphasize the primary purpose of outdoor radon measurements (as a tracer for the atmosphere). Additionally, it's important to highlight that there are ongoing atmospheric radon measurements at the ICOS stations and how these measurements would complement and benefit from the new emanation sources.*

**Line 23: …**and part of the *(uranium)* $^{238}$U-decay chain.

**Line 25:** "Approximately 3 % to 14 % of all lung cancer deaths are attributed to the exposure of radiation from 222Rn (progenies), depending on the activity concentration of 222Rn in a certain area" *– please add a reference.*

**Line 29:** *"*Additionally, the identification of RPAs is a major aim of the EU EMPIR project 19ENV01" **suggestion:** *Identification of the RPAs is a one of key objective of the EU EMPIR*

**Line 30-33:** "The project results will be implemented to identify RPAs, which is required by European Council Directive 2013/59/EURATOM, which in turn will help decision makers to enforce the respective 222Rn action plans of the EU member states and improve radiation protection of the general public (Röttger et al., 2021)."

**Suggestions:** *The project's outcomes will be utilized to fulfil the requirement set by the European Council Directive 2013/59/EURATOM, thereby enabling decision-makers to enforce $^{222}$Rn action plans within EU member states and enhance radiation protection for the general public (Röttger et al., 2021).*

**Line 34-35:** *"*In the framework of the International Carbon Observation System (ICOS) the European commission Joint Research Center (JRC) has published a map of Europe, presenting indoor $_{222}$Rn measurements as 35 early as 2006 (accessible at https://remap.jrc.ec.europa.eu/ Atlas.aspx# *accessed…* )." *– What is the connection between ICOS and radioactivity monitoring? Is this a typo?*
*Line 56-57: "*Integrated Radon Source Detector (IRSD)…" - from whom is developed this source? Since you have mentioned that CMI is developed from Czech…

**2. Measurements at PTB, Set up & Results**

**I would need a bit more clarification here:**

- *The Figure 1 (is IRSD #1 and RRI#1 in diffusion mode and 50L) and Figure 2 (IRSD #2; RRI#2 and 500 L in what type: diffusion/flow?) I'm particularly confused by the denser point measurements in Figure 1 compared to Figure 2. Could you please provide more clarity on this matter?*

- *Could you please ensure that the captions for Figure 1 and Figure 2 are consistent and harmonized? Perhaps you could consider including a legend in both figures to explain what blue line and grey points means, as well as indicating the specific experiment they represent?"*

- *Is there any difference between IRSD #1 and #2. Additionally, would it be more coherent to include Tables 1 and 2 in Section 2 for easier understanding? Alternatively, placing a separate section at the beginning of section 2 with a brief description of the sources (as it is in 3.4) in my opinion might enhance clarity.*

---

## Author Response (AR1)

**Letters to referees**

Referee Ileana Radulescu

Dear Mrs. Radulescu,

As the contact author I would like to answer to your comments.

*The manuscript is very well organized, the introduction is well demonstrated and support of an idea and the aim of the paper.*

Answer: Thank you very much for your support.

*I do have one minor question:*

*In section - 3.4 Integrated Radon Source Detector (IRSD) - about which AlphaGuard is considered? Is a different one from RRI #1 - RRI #4?*

Answer: It is RRI #3. I changed "AlphaGuard" to "RRI #3" in section 3.4 and added "#3" in table 4 of the revised manuscript.

*Question: If k (calibration factor is obtain for RRI #1 should be called k1, and so on for the rest). Is it the same factor? Maybe this is not clear , and it should be stated.*

Answer: I have attached numbers to all calibration factors corresponding to a specific RRI in the revised manuscript.

Thank you very much,

Tanita Ballé

This answer has been agreed to by all co-authors.

Anonymous referee

Dear anonymous referee,
As the contact author I would like to answer to your comments.

The study "**Two new 222Rn emanation sources – a comparison study" by Balle et al.** presents a comparison of two new 222Rn emanation sources using identical reference instrument at distinct sites: PTB and CMI. The findings revealed varying calibration factors, although they fall well within the targeted uncertainty of 10% (k=1). These sources play a crucial role in addressing a significant gap in traceability for both the radiation protection and climate monitoring communities. They demonstrate the fundamental capability to measure 222Rn activity concentrations below the 100 Bq m-3, with uncertainty of 10% (k=1). Achieving these uncertainties contributes on establishment of traceability to the International System of Units (SI).

Answer: Thank you very much for your support.

**My comments are below:**
**Introduction:** *The introduction is well-written; however, I recommend emphasizing the significance of these two sources for both the radiation protection and climate communities. While the importance in terms of radiation protection is evident, it's essential to also clarify its relevance to the climate community, which is currently missing.*
Answer: Thank you for this helpful remark. I have added a new paragraph to the introduction in the revised manuscript: "Aside from the radiation protection community precise outdoor $^{222}$Rn activity concentration measurements are also of great importance for the climate community. Levin et al. showed already in 1999 that $^{222}$Rn exhalation from soil can be used as a tracer to measure greenhouse gas emissions from soil, implementing the so-called Radon tracer method (Levin et al., 1999). For this reason, atmospheric $^{222}$Rn measurements are also carried out at stations of the International Carbon Observation System (ICOS)."

*Also, it is missing a short description or reference on current calibration sources.*
Answer: Besides the two types of sources presented in the paper there is, as far as I know, only one other type of $^{222}$Rn emanation source: The Pylon sources. I have included them in the revised manuscript: "Typical methods for the calibration of instruments use sources of $^{222}$Rn to create atmospheres of well-defined $^{222}$Rn activity concentration. Such sources are usually solid Pylon sources (Radulescu et al., 2022)."

**Line 45:** "222Rn activity concentration in air depends on a multitude of factors, like the Uranium concentration in soil, the temperature and soil permeability…" – *I would say that atmospheric radon concentration depends mostly on the atmospheric process (ref: Chambers et al., Kikaj et al.,)*
Answer: I have adjusted this sentence in the revised manuscript: "$^{222}$Rn activity concentration in air depends on a multitude of factors. Major factors include atmospheric processes like wind speed and temperature but also soil properties, like the Uranium concentration in soil and soil permeability, to name but a few (Čeliković et al., 2022)." Looking into the cited reference once more revealed that it is also suitable for the new part of the sentence.

**Line 41-48:** *I would restructure this section to emphasize the primary purpose of outdoor radon measurements (as a tracer for the atmosphere). Additionally, it's important to highlight that there are ongoing atmospheric radon measurements at the ICOS stations and how these measurements would complement and benefit from the new emanation sources.*
Answer: I have implemented this in the new paragraph. See above.

**Line 23: …**and part of the *(uranium)* 238U-decay chain.
Answer: I have implemented that in the revised manuscript: "… and part of the uranium ($^{238}$U)-decay chain."

**Line 25:** "Approximately 3 % to 14 % of all lung cancer deaths are attributed to the exposure of radiation from 222Rn (progenies), depending on the activity concentration of 222Rn in a certain area" – *please add a reference.*
Answer: The reference I found was more up-to-date than the old one so the numbers have changed a little: "Approximately 3 % to 12 % of all lung cancer deaths are attributed to the exposure of radiation

from $^{222}$Rn (progenies), depending on the activity concentration of $^{222}$Rn in a certain area (Martin-Gisbert et al., 2023)."

**Line 29:** *"Additionally, the identification of RPAs is a major aim of the EU EMPIR project 19ENV01"*
**suggestion:** *Identification of the RPAs is a one of key objective of the EU EMPIR*
Answer: The revised manuscripts reads like this: "… of RPAs is one of the key objectives of the EU EMPIR project …"

**Line 30-33:** "The project results will be implemented to identify RPAs, which is required by European Council Directive 2013/59/EURATOM, which in turn will help decision makers to enforce the respective 222Rn action plans of the EU member states and improve radiation protection of the general public (Röttger et al., 2021)." **Suggestions:** *The project's outcomes will be utilized to fulfil the requirement set by the European Council Directive 2013/59/EURATOM, thereby enabling decision-makers to enforce 222Rn action plans within EU member states and enhance radiation protection for the general public (Röttger et al., 2021).*
Answer: The revised manuscript reads like this: "The project outcomes will be utilized to fulfill the requirements set by European Council Directive 2013/59/EURATOM, thereby enabling decision makers to enforce the respective $^{222}$Rn action plans within the EU member states and enhance radiation protection for the general public (Röttger et al., 2021)."

**Line 34-35:** *"In the framework of the International Carbon Observation System (ICOS) the European commission Joint Research Center (JRC) has published a map of Europe, presenting indoor 222Rn measurements as 35 early as 2006 (accessible at https://remap.jrc.ec.europa.eu/ Atlas.aspx# accessed… )." – What is the connection between ICOS and radioactivity monitoring? Is this a typo?*
Answer: The connection is the radon tracer method. Still, to avoid confusion I removed the first half of the sentence and appended the second half at the end of the following sentence. It now reads like this: "All European countries operate automatic gamma dose rate systems and atmospheric radionuclide concentration detectors for environmental radioactive monitoring. The results of this radiological monitoring are exchanged through the European Radiological Data Exchange Platform (EURDEP) as requested by EU legislation (Council decision 87/600/Euratom of December 1987 on community arrangements for the early exchange of information in the event of radiological emergency ELI; Available at https://eur-lex.europa.eu/legal-content; accessed 21 April 2023) and the European commission Joint Research Center (JRC) has published a map of Europe, presenting indoor $^{222}$Rn measurements as early as 2006 (accessible at https://remap.jrc.ec.europa.eu/ Atlas.aspx#)."
This provides a great bridge to the next paragraph, so I thank you for the suggestion.

*Line 56-57:* "Integrated Radon Source Detector (IRSD)…" - from whom is developed this source? Since you have mentioned that CMI is developed from Czech…

Answer: I have added "developed by Physikalisch-Technische Bundesanstalt (PTB, Germany)" in the revised manuscript.

**2. Measurements at PTB, Set up & Results**
**I would need a bit more clarification here:**
*- The Figure 1 (is IRSD #1 and RRI#1 in diffusion mode and 50L) and Figure 2 (IRSD #2; RRI#2 and 500 L in what type: diffusion/flow?)*

Answer: It was operated in diffusion mode. To make that clear I have changed the first sentence of section 2.1 to "… (Radon Reference Instrument, RRI #1 and RRI #2, both of the AlphaGUARD EF type and operated in diffusion mode)."

*I'm particularly confused by the denser point measurements in Figure 1 compared to Figure 2. Could you please provide more clarity on this matter?*
Answer: I have added "The difference in point density between the two Figures is attributed to the difference in measurement time (more than 2 months in Figure 1 and less than 1 month in Figure 2) resulting in a higher number of measurements used in Figure 1." to section 2.3 of the revised manuscript. I hope this answers your question.

*- Could you please ensure that the captions for Figure 1 and Figure 2 are consistent and harmonized? Perhaps you could consider including a legend in both figures to explain what blue line and grey points means, as well as indicating the specific experiment they represent?"*
Answer: I did that on purpose to make it more interesting to read… But I changed it in the revised manuscript.

*- Is there any difference between IRSD #1 and #2. Additionally, would it be more coherent to include Tables 1 and 2 in Section 2 for easier understanding? Alternatively, placing a separate section at the beginning of section 2 with a brief description of the sources (as it is in 3.4) in my opinion might enhance clarity.*
Answer: The difference between the two IRSD is the amount of $^{226}$Ra they contain. I have added "As they were created with differing amounts of $^{226}$Ra they were expected to create atmospheres of differing $^{222}$Rn activity concentrations." in the revised manuscript.

Thank you very much,

Tanita Ballé

This answer has been agreed to by all co-authors.

---

## Referee Report (RR1)

[referee-annotated manuscript omitted]

---

## Author Response (AR2)

**Letters to referees**

Referee Ileana Radulescu

Dear Mrs. Radulescu,

As the contact author I would like to answer to your new comments.

"Approximately 3 % to 12 %" *delete 1*.

Answer: I have checked the reference (Martin-Gisbert et al., 2023) again and it says 12 %.

"As they were created with differing amounts of $^{226}$Ra they were expected to create atmospheres of differing $^{222}$Rn activity concentrations." *Because they were created with different amounts of $^{226}$Ra, they were expected to create atmospheres with different $^{222}$Rn activity concentrations.*

Answer: I have changed this in the revised manuscript.

Line 347 *Some how these calibration factors are not numbered in order, is k1, k2, then k5.*

Answer: After discussion with the co-authors it turned out that there was no 5$^{th}$ RRI and therefore I have changed "$k_5$" to "$k_3$" in the revised manuscript. Thank you for pointing out this error.

Thank you very much,

Tanita Ballé

This answer has been agreed to by all co-authors.